

# Spatiotemporal variability of oxygen concentration in coral reefs of Gorgona Island (Eastern Tropical Pacific) and its effect on the coral *Pocillopora capitata*

Ana Lucia Castrillón-Cifuentes[1,2], Fernando A. Zapata[2], Alan Giraldo[3] and Christian Wild[1]

[1] Department of Marine Ecology/Faculty of Biology and Chemistry, Universität Bremen, Bremen, Germany
[2] Departamento de Biología/Facultad de Ciencias Naturales y Exactas/Grupo de Investigación en Ecología de Arrecifes Coralinos, Universidad del Valle, Cali, Valle del Cauca, Colombia
[3] Departamento de Biología/Facultad de Ciencias Naturales y Exactas/Grupo de Investigación en Ciencias Oceanográficas, Universidad del Valle, Cali, Valle del Cauca, Colombia

Corresponding author
Ana Lucia Castrillón-Cifuentes,
analucia@uni-bremen.de,
castrillon.ana@correounivalle.edu.co

## ABSTRACT

Dissolved oxygen concentration (DO) is one of the main factors limiting benthic species distribution. Due to ocean warming and eutrophication, the ocean is deoxygenating. In the Eastern Tropical Pacific (ETP), deep waters with low DO ($<1$ mg $L^{-1}$) may reach coral reefs, because upwelling will likely intensify due to climate change. To understand oxygen variability and its effects on corals, we characterize the Spatio-temporal changes of DO in coral reefs of Gorgona Island and calculate the critical oxygen tension ($P_{crit}$) to identify the DO concentration that could represent a hypoxic condition for *Pocillopora capitata*, one of the main reef-building species in the ETP. The mean ($\pm$SD) DO concentration in the coral reefs of Gorgona Island was $4.6 \pm 0.89$ mg $L^{-1}$. Low DO conditions were due to upwelling, but hypoxia ($<3.71$ mg $L^{-1}$, defined as a DO value 1 SD lower than the Mean) down to 3.0 mg $O_2$ $L^{-1}$ sporadically occurred at 10 m depth. The $P_{crit}$ of *P. capitata* was 3.7 mg $L^{-1}$ and lies close to the hypoxic condition recorded on coral reefs during the upwelling season at 10 m depth. At Gorgona Island oxygen conditions lower than 2.3 mg $L^{-1}$ occur at $>20$ m depth and coincide with the deepest bathymetric distribution of scattered colonies of *Pocillopora*. Because DO concentrations in coral reefs of Gorgona Island were comparably low to other coral reefs in the Eastern Tropical Pacific, and the hypoxic threshold of *P. capitata* was close to the minimum DO record on reefs, hypoxic events could represent a threat if conditions that promote eutrophication (and consequently hypoxia) increase.

## INTRODUCTION

Oxygen ($O_2$) concentration is one of the main environmental factors limiting the occurrence of species in nature (*Dodds et al., 2007*). This is because, through metabolism, ingested food, and stored reserves are converted into energy to fuel any function in organisms (*Claireaux & Chabot, 2016*). In particular, $O_2$ works as the terminal electron acceptor in aerobic energy (ATP) production (*Wang et al., 2014*). The term normoxia

refers to a range of dissolved oxygen (DO) concentrations mostly observed in nature, while hypoxia to a level below the normoxic range that can trigger negative effects on organisms (*Welker et al., 2013*). The metabolic rate ($MO_2$) is one of the most common metrics used to assess the physiological performance of an organism under a particular ambient condition, and respirometry is a cost-effective way to measure it (*Harianto, Carey & Byrne, 2019*). Also, together with quantifying hypoxia tolerance, it is possible to make predictions about species' responses and resilience to ocean deoxygenation (*Negrete Jr & Esbaugh, 2019*; *Seibel et al., 2021*). Identifying the hypoxic threshold in aquatic organisms is a widely employed tool that helps to define the DO concentration from which organisms cannot sustain their physiological functions normally (*Hughes et al., 2020*). This oxygen limit is highly variable and depends on the taxa, life stage, exposure time, temperature, and previous $O_2$ history (*Vaquer-Sunyer & Duarte, 2008*; *Hughes et al., 2020*).

Classically, organisms are assigned to one out of two mechanisms for the regulation of their $MO_2$ when facing changes in $O_2$ conditions. To define the mechanism, $MO_2$ is plotted as a function of the environmental oxygen levels ($PO_2$); in a classic conformer, $MO_2$ declines in direct proportion to declining $PO_2$, and for an oxyregulator, the $MO_2$ remains constant down to a $PO_2$ level (called the critical oxygen tension), from which the $MO_2$ conforms to the environmental $O_2$ condition, but with an energetic cost for the organism (*Regan et al., 2019*; *Claireaux & Chabot, 2016*; *Cobbs & Alexander Jr, 2018*; *Negrete Jr & Esbaugh, 2019*).

However, organisms rarely perform as strict oxyregulators, and instead, its a spectrum of responses has been observed; the regulatory ability and the regulatory index are similar metrics that quantify an animal's ability to regulate $MO_2$ in response to changes in $O_2$ conditions (*Cobbs & Alexander Jr, 2018*; *Mueller & Seymour, 2011*). Other common methods to assess hypoxia tolerance include measuring the time to loss of equilibrium, which is the amount of time an animal can survive when forced to rely on anaerobic metabolism (*Negrete Jr & Esbaugh, 2019*), and determining the critical oxygen tension ($P_{crit}$).

## Effects of hypoxia on scleractinian corals

Hypoxia has been shown to promote bleaching in *Acropora nobilis* and *Alveopora verrilliana* (*Baohua et al., 2004*). *Montipora peltiformis* was able to tolerate anoxia (absence of $O_2$) for up to 4 days, but its co-occurrence with acidification becomes lethal within 24 h (*Weber et al., 2012*). Additional to bleaching, hypoxia ($<4$ mg $L^{-1}$) has been shown to lead to tissue loss, increased respiration rate, and reduced photosynthetic $O_2$ production in *Acropora yongei* (*Haas et al., 2014*). *Acropora cervicornis* has been observed to experience tissue loss after one day of exposure to 1.0 mg $L^{-1}$ of $O_2$, and mortality after five days; however, in *Orbicella faveolata* the same conditions had no effect after a week (*Johnson et al., 2021*). *Gravinese et al. (2021)* found that *O. faveolata* significantly reduced their respiration rate (34.2% and 62.8%) when exposed (1 h) to hypoxia (0.77 mg $L^{-1}$) or hypoxia and warming (31.4 °C). *Pocillopora*, *Acropora* and *Motipora* corals suffered mass (71%) mortality (as evidenced by tissue peeling without decoloration) when DO was between $1.4 - 2.0$ mg $L^{-1}$ due to a micro-algal bloom (*Raj et al., 2020*).

Compared to pH conditions, $O_2$ also has a great effect on calcification. In healthy *Montastraea faveolota*, an increase in DO (from 5.4 to 7.8 mg $L^{-1}$) promotes dark calcification, but in bleached corals, both glycerol and well-oxygenated conditions were required to continue calcification (*Colombo-Pallotta, Rodríguez-Román & Iglesias-Prieto, 2010*). The optimal $O_2$ condition for calcification in *Galaxea fascicularis* was 7.3 mg $L^{-1}$, while 3.3 mg $L^{-1}$ (under light or dark conditions) reduced calcification even in fed corals (*Wijgerde et al., 2012*). Likewise, 1.9 mg $L^{-1}$ of $O_2$ at night decreased (51%) calcification in *Acropora millepora* regardless of the pH condition (*Wijgerde et al., 2014*).

In *Symbiodinium microadriaticum*, hypoxia (6% $O_2$ saturation) reduces the activity of Superoxide dismutase, Catalase, and Ascorbate peroxidase (*Matta & Trench, 1991*). Isolated zooxanthellae from *Dichocoenia stokesii* reduced their photosynthesis and respiration rates at 50% $O_2$ saturation (*Gardella & Edmunds, 1999*), and 0% air saturation reduced the photochemical efficiency of PSII in zooxanthellae extracted from *Pocillopora damicornis* (*Ulstrup, Hill & Railp, 2005*). However, hypoxia (20% $O_2$ saturation) had no effects on zooxanthellae of *Galaxea fascicularis* (*Osinga et al., 2017*).

Anaerobic metabolism was the fast response of *Montipora capitata* to hypoxia during diel changes in $O_2$ (hyperoxia during the day, due to photosynthesis, and hypoxia at night, result of respiration); however, this response was inefficient for survival under prolonged hypoxia (5 days) (*Murphy & Richmond, 2016*). The Hypoxia Inducible Factor (HIF-1) was responsible for maintaining $O_2$ homeostasis in *Stylophora pistillata* and *Acropora tenuis* (*Zoccola et al., 2017*; *Alderdice et al., 2020*). In *Acropora yongei*, metabolic enzymes are expressed constantly throughout the diel cycle, with only Strombine (a fermentation end-product) peaking at the onset of hypoxia and hyperoxia (*Linsmayer et al., 2020*). Larvae of *Acropora selago*, when exposed to 12 h of hypoxia at night and 12 h of oxygenation during the day, experienced negative effects on pathways related to Homeobox genes, mitochondrial activity, and lipid metabolism (*Alderdice et al., 2021*). In *Acropora* spp, hypoxia (1.75 mg $L^{-1}$) during the night causes oxidative stress and DNA damage, without activation of the antioxidant defense system (*Deleja et al., 2022*).

## Ocean deoxygenation

Ocean warming and acidification are considered leading contributors to live coral cover decline on coral reefs around the world (*Hoegh-Guldberg et al., 2017*; *Heron et al., 2017*; *Hughes et al., 2018*). However, growing evidence indicates that ocean deoxygenation (due to low solubility of $O_2$ in a warmer ocean, and eutrophication) is another immediate threat to corals' survival (*Altieri et al., 2017*; *Nelson & Altieri, 2019*; *Hughes et al., 2020*). In the last 50 years, the ocean has lost 2% of its $O_2$ (*Schmidtko, Stramma & Visbeck, 2017*), and this loss is expected to climb to 5% by 2100 under the Representative Concentration Pathway (RCP) 8.5 (*Keeling, Körtzinger & Gruber, 2010*), a high green-house gases emission ("business as usual") scenario, or a Shared Socio-economic Pathways, SSP5, (*Ho et al., 2019*).

Currently, 13% of tropical coral reefs are at risk of hypoxia (*Altieri et al., 2017*). A meta-analysis by *Sampaio et al. (2021)* demonstrated that hypoxia has more negative effects on marine animals (including corals) than ocean warming and acidification. During

the End-Permian period, ocean warming and deoxygenation, rather than ocean warming and acidification, triggered the mass extinction of ancient reefs (*Penn et al., 2018*).

In the Eastern Tropical Pacific region (ETP) the oxygen minimum zones ($<1\,\text{mg}\,\text{L}^{-1}$) are expanding vertically, and there is a trend of oxygen loss of $49\,\text{mmol}\,\text{m}^{-2}\,\text{year}^{-1}$ (*Stramma et al., 2008*). Additionally, climate change will cause increased stratification and upwelling of hypoxic waters onto the surface layer, leading to adverse effects for benthic organisms, including coral reefs (*Fiedler & Lavín, 2017*). Coastal coral reefs can also experience hypoxic conditions when water quality decreases because of land-based runoff (*Altieri et al., 2017*; *Kealoha et al., 2020*). The ETP region is recognized for hosting a few species of scleractinian corals that live under extreme environmental conditions (high $pCO_2$, low aragonite saturation, high nutrients, fluctuating temperatures, and intense bioerosion, *Glynn, Manzello & Enochs, 2017*). Hence, additional stress from climate change, including particularly low DO, imposes a serious threat to the coral reefs in this region.

As expected from the environmental conditions in the ETP, coral reefs of Gorgona Island (Colombian Pacific) develop under particularly limiting conditions: they occur in waters with one of the lowest salinities for reef development in the world (*Kleypas, McManus & Meñez, 1999*; *Blanco, 2009*; *Guan, Hohn & Merico, 2015*), and the water temperature varies widely from as low as 16 °C during upwelling events to up to 32 °C during ENSO events (*Vargas-Ángel et al., 2001*; *Giraldo, Rodríguez-Rubio & Zapata, 2008*; *Zapata, 2017*). However, despite the low diversity (24 species; *Glynn, Manzello & Enochs, 2017*), corals grow vigorously, particularly those in the genus *Pocillopora* (mean linear extension from 1.89 to 4.08 cm year$^{-1}$; *Lizcano-Sandoval, Londoño-Cruz & Zapata, 2018*; *Céspedes-Rodríguez & Londoño-Cruz, 2021*) and reefs exhibit relatively high live coral cover (50.7 $\pm$ 5.7%, mean $\pm$ SE; *Zapata, 2017*). Here, coral reefs seem to be quite resilient because they fully recovered in less than a decade from the 1982-83 El Niño mass bleaching and mortality event, have not been significantly affected by more recent El Niño events, and have recovered in a relatively short time from significantly detrimental sub-aerial exposure events (*Vargas-Ángel et al., 2001*; *Navas-Camacho, Rodríguez-Ramírez & Reyes-Nivia, 2010*; *Zapata et al., 2010*; *Glynn, Manzello & Enochs, 2017*; *Zapata, 2017*).

Coral reefs at Gorgona Island (and in the ETP in general) have been studied for longer than three decades, but information about corals' physiological thresholds in response to stressors is scarce. Furthermore, there are limited descriptions of the micro-environment (*i.e.*, at each reef, <1 km) with which corals cope. Given the effect of global climate change and local stressors on DO availability and its effects on the survivorship of scleractinian corals, this research aims to, first, characterize the spatial (reefs and depths) and temporal (seasons, and years) variability of DO concentration in areas where scleractinian corals abound around Gorgona Island and, second, to identify the DO concentration that represents a hypoxic condition for *Pocillopora capitata* corals, one of the main reef-building corals in the ETP region.
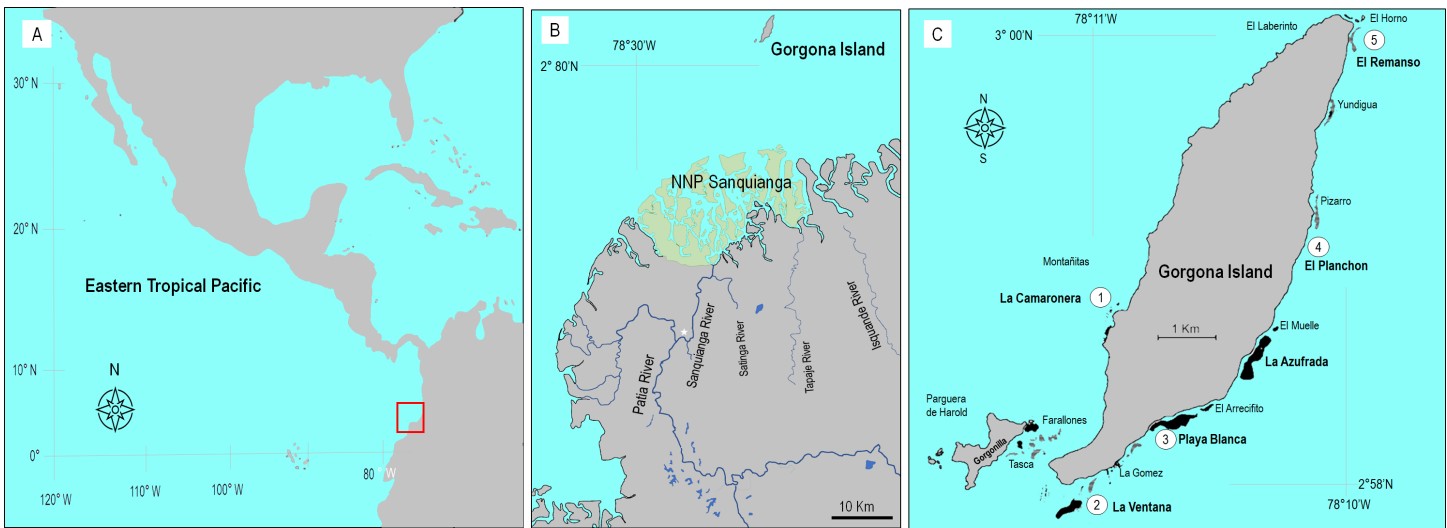

**Figure 1** **Location of Gorgona Island.** (A) The red square indicates the location of Gorgona Island in the Eastern Tropical Pacific. (B) Position of Gorgona Island in front of the mangrove ecosystem at the National Natural Park (NNP) Sanquianga (green area), where sediments from the Patia River are discharged into the ocean. The white star indicates the position of a constructed channel that connects the Sanquianga and Patia rivers. (C) Numbers (from 1 to 5) indicate the location of coral reefs where dissolved oxygen was monitored.

## MATERIALS & METHODS

### Characterization of dissolved oxygen concentrations in coral reefs of Gorgona Island

Gorgona (2°58′10″N–78°11′05″W, Fig. 1) is a continental island in the Colombian Pacific, 9.3 km long and 2.6 km wide, located 30 km off from the nearest point on the continent (*Giraldo, Rodríguez-Rubio & Zapata, 2008*; *Muñoz, Jaramillo-González & Zapata, 2018*), and in front of the Sanquianga National Natural Park where high sediment discharge from the Patia River occurs (*Restrepo & Cantera, 2013*). This island is a natural national park and hosts several well-developed (~1 km long, 8 m thick) coral reefs, incipient and relict reefs, and several coral communities (*Glynn, Prahl & Guhl, 1982*; *Zapata & Vargas-Ángel, 2003*).

To determine the oxygen conditions in which corals cope at reefs, DO concentration (in mg L$^{-1}$) data were extracted from the technical reports of the environmental monitoring program of Gorgona Island (a list of the reports is in SM1) implemented by the Oceanographic Sciences research group from Universidad del Valle (Cali, Colombia). Since 2005 the program assessed marine environmental conditions during a week in the first and the second semester of each year (March or April, for the upwelling season, and September, October, or November, for the non-upwelling season), and in five monitoring stations around the island, that are close to coral reefs (Fig. 1). At each monitor station, and during daylight hours, a water sample was collected (at 1 and 10 m depth) with a Niskin bottle (5 L). And a YSI-85 (YSI Inc) was employed to measure DO.

The YSI-85 is a handheld multiparameter sensor (polarographic), that measures DO, conductivity, salinity, and temperature. Before each oceanographic campaign, the

membrane and the 0 $O_2$ electrolyte solution of the sensor were replaced. Then, the sensor was calibrated according to the manufacturer's instructions. In short, three to six drops of clean water were added to a sponge located in an internal chamber of the sensor; this creates a 100% water-saturated air environment for dissolved oxygen calibration. The calibration of the sensor was checked in a 0 and 100% air satured solution. Also, measurements were calibrated according to the altitude of the sampling sites (0 msl, 760 mmHg). The oxygen sensor exhibited a range of measurements from 0 to 20 mg $L^{-1}$, a resolution of 0.01 mg $L^{-1}$, and an accuracy of $\pm 0.3$ mg $L^{-1}$. The measurement of DO was done inside the Niskin bottle once it arrived on board. When the bottle was opened, the probe was introduced (making slow circular movements) and the DO data was recorded when a DO value was stabilized.

The monitoring stations (Fig. 1C) correspond to La Ventana reef, Playa Blanca reef, El Remanso reef, La Camaronera reef, and El Planchon (an artificial reef). The available data included the years 2005, 2006, 2008, 2009, 2012–2014, and 2016–2019. The full data set was highly imbalanced since it did not include complete data for both seasons or depths in most years. Hence, to rigorously examine the effects of temporal and spatial factors on DO variability, we used the data from the years 2013, 2017, and 2019, which were complete for all levels of the four factors.

A factorial nested ANOVA was performed using the General Linear Models module of Statistica software (Statsoft®) to establish the effects on DO of the categorical factors Reef (5 levels) and Depth (2 levels, nested within Reef) crossed with Year (3 levels) and Season (2 levels, nested within Year). Assumptions of homogeneity of variances and normal distribution of residuals were examined with a Levene's Test ($F = 1.18$, $df = 9,50$, $p = 0.32$) and a Kolmogorov–Smirnov test with Lilliefors correction ($D = 0.113$, $p = 0.06$), respectively.

We follow the definition of Welker et al. (2013) to establish the normoxic and hypoxic conditions at Gorgona's reefs. Here we defined normoxia (and the normoxic range) according to the DO concentration values within the Mean $\pm$ 1 SD, while hypoxia as any value of DO below the Mean minus 1 SD.

Aiming to describe DO conditions, particularly at different depths, at La Azufrada reef (where coral samples were collected for the determination of the coral hypoxic threshold), additional DO measurements were done with a YSI-85 on March 2 and 5, 2022 on the reef or its immediate vicinity offshore. Eighteen water samples were taken at the surface (1 m) and seven water samples were taken at 10 m depth. Mean DO values for each depth were compared with a two-sample $t$-test with separate variance estimates due to significant variance heterogeneity (Levene's test, $F = 8.27$, $df = 23$, $p = 0.008$).

## Identification of the hypoxic threshold

Coral reefs of Gorgona Island (and in general in the ETP) are dominated by *Pocillopora* corals (Glynn, Manzello & Enochs, 2017). However, the morphological and molecular characteristics within this genus in the region do not yet allow to clearly delimit species (Pinzón et al., 2013). For this study, we selected colonies that follow the morphological

features of *Pocillopora capitata* (*Veron, 2000*), one of the most common species at Gorgona Island.

In November 2021, seven coral colonies of *P. capitata* were selected from the outer reef slope (∼7 m depth) of La Azufrada reef. A fragment of each one was collected and transported to the wet lab facility of the Henry von Prahl Research Station of Gorgona Island (located 1.1 km from La Azufrada reef, at 0 msl, 760 mmHg). Coral fragments were kept in a 70 L tank (filled with water collected from the reef), in which water movement was generated by a submersible pump. Coral fragments were maintained in dark conditions for 2 h before the experiment.

To describe the relationship between the respiration rate of *P. capitata* and the environmental oxygen availability, we incubated coral fragments in sealed respiration chambers to allow DO depletion due to respiration by the corals. The resultant recordings of DO inside the chambers were then used to calculate the corals' $MO_2$ and relate it with a gradient of environmental oxygen saturation to identify the $P_{crit}$.

The experimental setup (Fig. 2) consisted of eight hermetically sealed respiration chambers (with internal movement of water generated by a peristaltic pump), seven containing a coral fragment, and one without coral (363 mL of water) to measure microbial respiration. All chambers were filled with seawater collected from the reef (30 PSU, 28 °C), and with an initial DO concentration of 6.7 mg $L^{-1}$ due to water movement with a submersible pump.

In the inner wall of the chambers, an oxygen sensor spot (OXSP5, PyroScience GmbH) was glued. DO inside each chamber was recorded every ∼20 min over the course of 5 h by manually locating an optical fiber (SPFIB-LNS, PyroScience GmbH) on the sensor spot (on the external side of the chamber). The optical fiber was connected to an oxygen meter that recorded the DO concentration adjusted to salinity, temperature, and atmospheric pressure (Firesting-GO2, PyroScience GmbH).

Following the oxygen meter manufacturer's instructions, oxygen-free water, and 100% $O_2$-saturated water were used to calibrate the oxygen sensors. Oxygen-free water was prepared using 1.5 g of sodium sulfite ($Na_2SO_3$) powder dissolved in 50 mL of deionized water while stirring. For the 100% $O_2$-saturated water, air was pumped into 50 mL of water while stirring using an aquarium air pump. After 20 min, the air pump was switched off and the solution was stirred for another 10 min.

Data recording for oxygen consumption started at 6.7 mg $L^{-1}$ (101% air saturation), approximately 1 h after closing the chambers to allow acclimation of corals to the closed system. Measurements were done at 30 PSU, 28 ± 0.3 °C (mean ± SD) and in the dark, by placing all respiration chambers in a water bath (within an aquarium) containing a thermostat and a submersible water pump to maintain a constant temperature throughout the aquarium and all chambers incubated there. As stated previously, data recording stopped after 5 h, at which point the DO inside the respiration chambers was <2.0 mg $L^{-1}$, a value considered as hypoxic (*Vaquer-Sunyer & Duarte, 2008*), and lower than the recorded at coral reefs of Gorgona Island (see results).

We selected the critical oxygen tension ($P_{crit}$) metric to identify the DO concentration that represents a hypoxic condition for *P. capitata* corals. $P_{crit}$ was defined by *Regan*
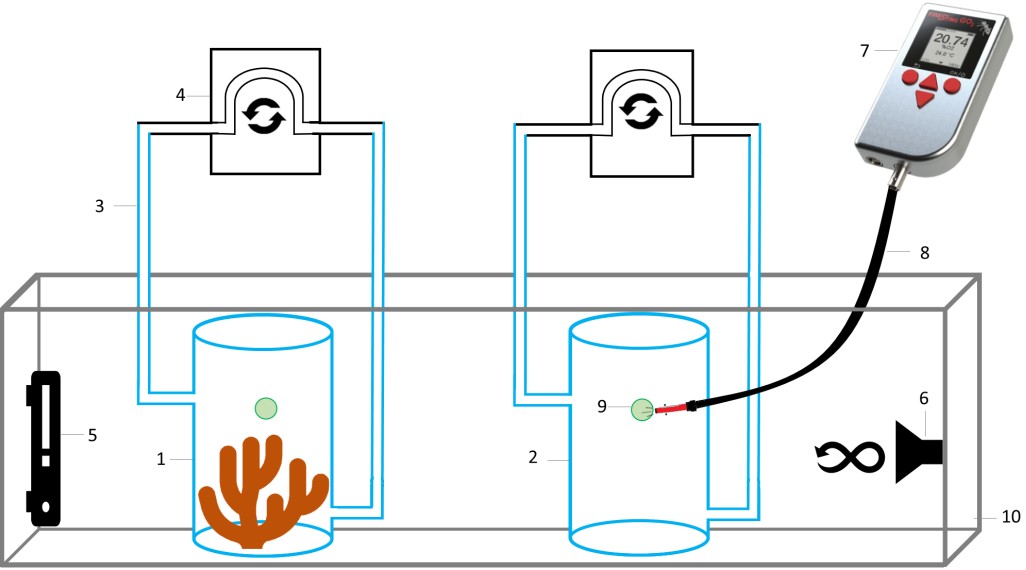

**Figure 2 Schematic representation of the experimental setup to measure oxygen consumption by *Pocillopora capitata*.** (1) Respiration chamber with a coral fragment, (2) control respiration chamber for microbial respiration, (3) hoses for water movement, (4) peristaltic pump, (5) thermostat, (6) submersible water pump, (7) hand-held oxygen meter connected to an (8) optical fiber, to measure dissolved oxygen on each respiration chamber by placing it on the (9) oxygen sensor spot, (10) water bath at 28 °C.

*et al. (2019)* as "the lower bound of the oxygen partial pressure ($PO_2$) spectrum over which an animal supports its $MO_2$ predominantly using aerobic metabolism, albeit with a diminishing aerobic scope for activity as $PO_2$ approaches $P_{crit}$". From this point, the $MO_2$ conforms to the environmental $O_2$ condition, but with a cost for the organism, or until $PO_2$ conditions became lethal (*Claireaux & Chabot, 2016*; *Cobbs & Alexander Jr, 2018*; *Negrete Jr & Esbaugh, 2019*). From the $P_{crit}$ the organism is exposed to severe hypoxia (*Grieshaber et al., 1994*). $P_{crit}$ is considered the most appropriate benchmark when measured in fasted and resting animals (the standard $MO_2$) because it is truly 'critical' for the animal's survival (*Ultsch & Regan, 2019*).

$P_{crit}$ is considered a powerful tool to assess hypoxia tolerance, as a lower value is consistent with more tolerance (*Negrete Jr & Esbaugh, 2019*). Because the $P_{crit}$ is strongly correlated with the environmental $O_2$ level that organisms face, it is therefore ecologically relevant and allows to identify the detrimental $O_2$ ambient conditions (*Regan et al., 2019*).

We used the packages respR (http://cran.r-project.org/package=respR) and respirometry (http://cran.r-project.org/package=respirometry) to calculate corals' respiration rate and $P_{crit}$ (*Harianto, Carey & Byrne, 2019*; *Seibel et al., 2021*). The respR package used the time *versus* oxygen data to calculate whole organism $MO_2$ and $P_{crit}$ (methods: broken stick regression, and segmented regression, and we set width to 0.1 and 0.2 for its calculation). The respirometry package used the time *versus* oxygen data to calculate $MO_2$; to calculate $P_{crit}$ we used $MO_2$ *versus* $PO_2$ as input, and three methods for its quantification: $\alpha$-Pcrit,

broken stick regression, and nonlinear regression (NLR). Differences in the resulting $P_{crit}$ values were assessed between packages and methods (and widths).

The broken stick method depends on a relatively constant $MO_2$ as $PO_2$ declines, and a discontinuity in $MO_2$ must be taken as $P_{crit}$ (*Yeager & Ultsch, 1989*). The segmented method estimates the $P_{crit}$ by iteratively fitting two intersecting models and selecting the value that minimizes the difference between the fitted lines (*Muggeo, 2003*). For the nonlinear regression (NLR) the $P_{crit}$ is an inflection point in the data after being fit to different functions, and the best function was selected according to the smallest Akaike's Information Criterium (*Muggeo, 2003*). The Physiological oxygen supply capacity ($\alpha$) is the maximum amount of oxygen that can be supplied per unit of time and oxygen pressure, and $\alpha$-$P_{crit}$ is the $PO_2$ at which physiological oxygen supply mechanisms are operating at maximum capacity. $\alpha$ is a species- and temperature specific constant that describes the linear dependency of $P_{crit}$ on $MO_2$ (*Seibel et al., 2021*).

The $MO_2$ (mg $O_2$ h$^{-1}$) was standardized by the weight (g) of coral tissue (mg $O_2$ h$^{-1}$ g$^{-1}$), estimated by the difference between wet weight ($31.2 \pm 10.6$ g) and ash-free dry weight ($28.0 \pm 9.5$ g) of coral fragments.

The environmental authority of Parques Nacionales Naturales de Colombia approved all aims and methods of this research (PIR NO.014.19, Agreement 239; December 20, 2019).

# RESULTS

## Temporal and spatial variability of dissolved oxygen concentration
The mean DO concentration on reefs ranged from 4.3 to 4.9 mg L$^{-1}$ (Fig. 3A). DO at surface was $4.5 \pm 0.7$ mg L$^{-1}$ (mean $\pm$ SD), and at 10 m was $4.6 \pm 1.0$ mg L$^{-1}$ (Fig. 3B). According to the Nested Anova (Table 1), there were no differences in DO concentration between reefs or depths at each reef, but the factors Year (Fig. 3C) and season (Fig. 3D) had a significant effect on DO variability. In 2019, DO concentration was significantly higher and more variable ($5.3 \pm 1.0$ mg L$^{-1}$) than in 2017 ($4.3 \pm 0.3$ mg L$^{-1}$) or 2013 ($4.1 \pm 0.6$ mg L$^{-1}$). During the upwelling season DO concentration was lower than during the non-upwelling season, and marked differences were recorded in 2013 and 2017 (Table 1). Regarding the interaction between depth and season on DO concentration (Fig. 3E), significant lower values ($4.0 \pm 0.9$ mg L$^{-1}$) occurred during the upwelling season at 10 m depth (Fig. 3E. Two-way Anova (Table 1); Tukey HSD = 0.6, *df* = 56, *p* < 0.04).

## Normoxic range and hypoxic conditions
The range of normoxic conditions on coral reefs of Gorgona Island was established based on the average value of DO: $4.6 \pm 0.89$ mg L$^{-1}$, and values below 3.71 mg L$^{-1}$ (DO < mean $-$ SD) should be considered hypoxic. Such conditions, with values between 3.70 and 3.0 mg L$^{-1}$ (lowest DO reported), were recorded in 2013 and 2017 during the upwelling season at 10 m depth, and represented 16% of all data points.

## DO concentration at La Azufrada reef
There were significant differences in DO concentrations, during March 2022, between the surface and 10 m water depth (Student's *t* test = 5.7, *df* = 7.17, *p* = 0.0006). In the

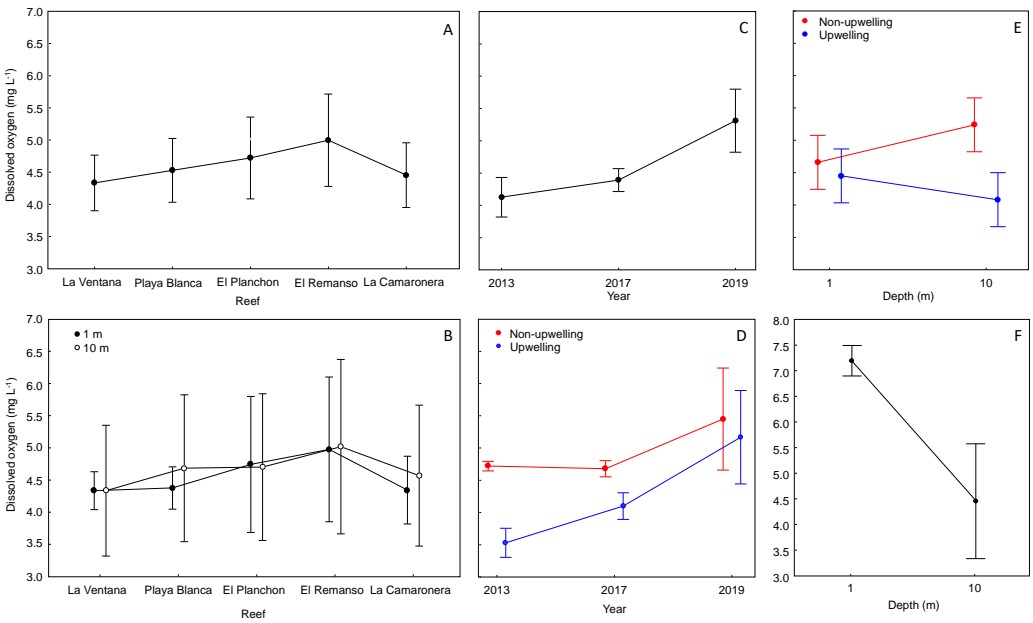

**Figure 3** **Effect of temporal and spatial factors on dissolved oxygen concentration (DO) at reefs of Gorgona Island.** (A) DO concentration in reefs of Gorgona Island. (B) DO concentration at 1 and 10 m depth in reefs of Gorgona Island. (C) Interannual variation in DO. (D) Seasonal variation of DO between years, red corresponds to the non-upwelling season and blue to the upwelling season. (E) DO concentration at 1 and 10 m depth between seasons. (F) DO concentration at 1 and 10 m depth in La Azufrada reef during the upwelling season of 2022. In all panels: mean (●), and 0.95 confidence intervals (I).

surface water layer, the mean DO concentration was $7.3 \pm 0.5$ mg L$^{-1}$, and minimum and maximum values were $6.0$ mg L$^{-1}$ and $8.4$ mg L$^{-1}$. At 10 m depth, the mean DO was $4.8 \pm 1.1$ mg L$^{-1}$, and minimum ($3.1$ mg L$^{-1}$) and maximum ($5.8$ mg L$^{-1}$) values were lower than at the surface (Fig. 3F).

## Hypoxic threshold of *Pocillopora capitata*

The mean ($\pm$ SD) MO$_2$ of *P. capitata* coral fragments at 28 °C was $0.26 \pm 0.02$ mg O$_2$ h$^{-1}$, and there were no differences in the results obtained from the two packages employed to calculate it (Table 2, Student's *t* test in Supplementary material 2).

The mean $P_{crit}$ calculated with the respR package was $3.5 \pm 0.9$ mg O$_2$ L$^{-1}$ (Fig. 4, Table 3), and according to a nested Anova, there were no statistical differences if a particular method was selected or if the width value was set to 0.1 or 0.2 (Supplementary material 3).

However, different $P_{crit}$ values were obtained when the respirometry package was employed (One-way Anova, *Sum of squares* $= 23.5$, *df* $= 2$, $F = 9.4$, $p = 0.001$. Assumptions: homogeneity of variance [Cochran test, $C = 0.5$, *df* $= 2$, $p = 0.3$]; normal distribution of residuals [$D = 0.14$, $p > 0.2$]). Both the broken stick regression and NLR methods produced similar results (Fig. 5, Table 3), but the $\alpha$-Pcrit method produced significantly lower values (Tukey HSD test $= 1.2$, *df* $= 18$, $p < 0.004$). This is because this method describes the oxygen limit for the MO$_2$ (*Seibel et al., 2021*).

**Table 1 Statistical analysis to examine differences in Dissolved oxygen concentration at coral reefs of Gorgona Island.**

| Test | Factor | SS | df | MS | F | p |
|------|--------|----|----|----|----|----|
| *Nested Anova* | Reef | 3.22 | 4 | 0.80 | 1.94 | 0.11 |
| | Depth (in Reef) | 0.44 | 5 | 0.08 | 0.21 | 0.95 |
| | Year | 15.44 | 2 | 7.72 | 18.64 | 0.000001 |
| | Season (in Year) | 9.16 | 3 | 3.05 | 7.37 | 0.0004 |
| | Error | 18.63 | 45 | 0.41 | | |
| *Factorial Anova* | Depth | 0.16 | 1 | 0.16 | 0.26 | 0.61 |
| | Season | 7.01 | 1 | 7.01 | 10.81 | 0.001 |
| | Depth × Season | 3.37 | 1 | 3.37 | 5.19 | 0.02 |
| | Error | 36.35 | 56 | 0.64 | | |

**Table 2 Metabolic rate ($MO_2$) of *Pocillopora capitata*.**

| Coral | respR | | | Respirometry | | |
|-------|-------|---|---|--------------|---|---|
| | $MO_2$ (mg $O_2$ $h^{-1}$) | $MO_2$ (mg $O_2$ $h^{-1}$ $g^{-1}$) | $R_2$ | $MO_2$ (mg $O_2$ $h^{-1}$) | $MO_2$ (mg $O_2$ $h^{-1}$ $g^{-1}$) | $R_2$ |
| A | 0.256 | 0.071 | 0.88 | 0.266 | 0.074 | 0.88 |
| B | 0.223 | 0.046 | 0.86 | 0.233 | 0.048 | 0.85 |
| C | 0.287 | 0.137 | 0.97 | 0.297 | 0.142 | 0.97 |
| D | 0.259 | 0.069 | 0.86 | 0.269 | 0.071 | 0.86 |
| E | 0.287 | 0.105 | 0.91 | 0.297 | 0.108 | 0.90 |
| F | 0.257 | 0.064 | 0.70 | 0.267 | 0.066 | 0.69 |
| G | 0.249 | 0.146 | 0.98 | 0.256 | 0.151 | 0.98 |
| **Mean ± SD** | 0.260 ± 0.02 | 0.091 ± 0.03 | 0.88 ± 0.09 | 0.269 ± 0.02 | 0.094 ± 0.03 | 0.88 ± 0.10 |

**Notes.**

$MO_2$ (mg $O_2 h^{-1}$ and mg $O_2$ $h^{-1}$ $g^{-1}$) of *Pocillopora capitata* fragments at 28 °C, 30 PSU, and dark conditions was calculated using two R packages: respR (*Harianto, Carey & Byrne, 2019*), and respirometry (*Seibel et al., 2021*). $R_2$ is the coefficient of determination for the linear regression fit to calculate $MO_2$.

**Table 3 Critical oxygen tension ($P_{crit}$) of *Pocillopora capitata*.**

| Coral | respR | | | | Respirometry | | | |
|-------|-------|---|---|---|--------------|---|---|---|
| | Broken stick | | Segmented | | | | | |
| | *0.1* | *0.2* | *0.1* | *0.2* | *α-Pcrit and α* | | *Broken stick* | *Nonlinear regression* |
| A | 3.6 | 3.2 | 4.0 | 2.8 | 1.4 | 23.87 | 4.3 | 4.2 |
| B | 3.3 | 3.3 | 3.6 | 2.8 | 2.9 | 15.83 | 4.3 | 4.9 |
| C | 5.4 | 3.7 | 4.9 | 5.1 | 2.4 | 6.24 | 5.0 | 4.1 |
| D | 2.6 | 2.5 | 3.1 | 2.0 | 0.9 | 25.56 | 3.0 | 3.8 |
| E | 2.7 | 2.9 | 4.2 | 3.7 | 1.1 | 18.88 | 4.4 | 0.7 |
| F | 3.7 | 3.1 | 3.90 | 2.8 | 1.4 | 33.00 | 3.1 | 4.7 |
| G | 5.0 | 4.6 | 5.37 | 2.3 | 0.8 | 8.76 | 2.1 | 4.5 |
| Mean ± SD | 3.8 ± 1.0 | 3.3 ± 0.6 | 4.1 ± 0.7 | 3.1 ± 1.0 | 1.5 ± 0.8 | 18.8 ± 9.4 | 3.7 ± 1.0 | 3.8 ± 1.4 |

**Notes.**

$P_{crit}$ (mg $O_2$ $L^{-1}$) of *Pocillopora capitata* fragments at 28 °C, 30 PSU, and dark conditions was calculated using two R package: respR (*Harianto, Carey & Byrne, 2019*), and respirometry (*Seibel et al., 2021*). In respR package, and for the broken stick (midponit) and Segmented methods we set the width at 0.1 and 0.2. From the respirometry package we calculated the oxygen supply capacity ($α$), which is the maximum amount of oxygen that can be supplied per unit of time and oxygen pressure ($μ$mol $O_2$ $g^{-1}$ $h^{-1}$ $kPa^{-1}$).

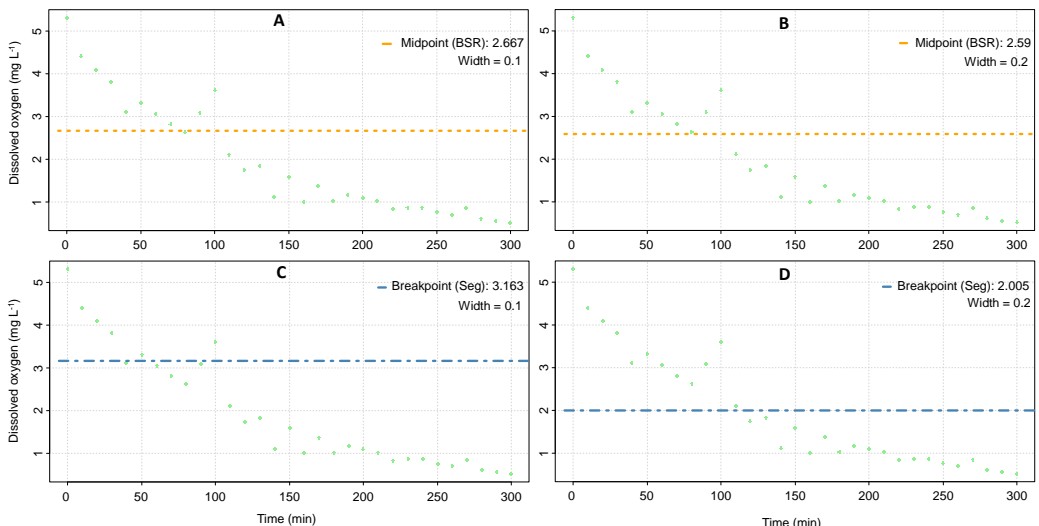

**Figure 4** **Critical oxygen tension ($P_{crit}$) of *Pocillopora capitata* calculated with the respR package.** $P_{crit}$ (mg $O_2$ $L^{-1}$) of a fragment of *Pocillopora capitata* (incubated at 28 °C, 30 PSU, and dark conditions) was calculated using the respR package (*Harianto, Carey & Byrne, 2019*). The panels show the $P_{crit}$ from the broken stick method with the width set at 0.1 (A) and 0.2 (B), and the $P_{crit}$ from the segmented method with width at 0.1 (C) and 0.2 (D).

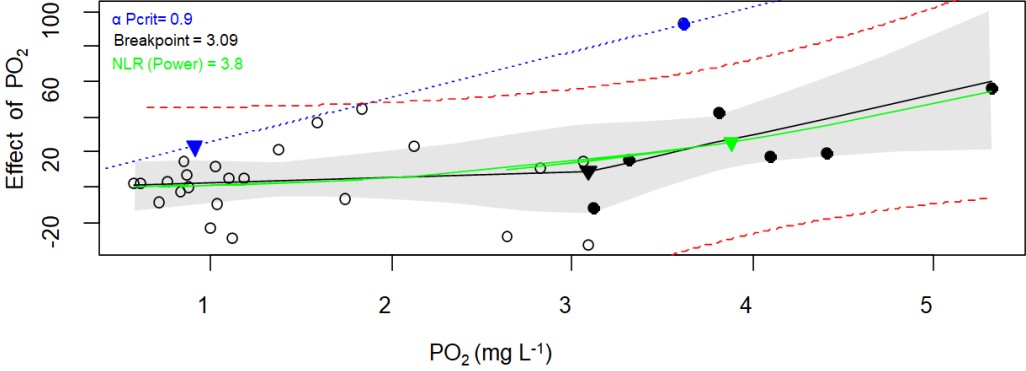

**Figure 5** **Critical oxygen tension ($P_{crit}$) of *Pocillopora capitata* calculated with the respirometry package.** $P_{crit}$ (mg $O_2$ $L^{-1}$) of a fragment of *Pocillopora capitata* (incubated at 28 °C, 30 PSU, and dark conditions) was calculated using the respirometry package (*Seibel et al., 2021*). Three $P_{crit}$ metrics were plotted: the broken stick regression (black line, $P_{crit}$ (▼) = 3.09), the nonlinear regression metric (green line, $P_{crit}$ (▼) = 3.8), and the $\alpha$-based Pcrit method (blue dash line, $\alpha$Pcrit (▼) = 0.9). Black circles represent oxyregulating observations, while empty circles represent the oxyconforming observations. For the Broken stick regression, the dashed red curves is the 95% prediction interval. The NLR curve was fitted to the Power function (selected for its smallest AIC from the Michaelis-Menten, Hyperbola, Pareto, and Weibull functions). The blue line represents $\alpha$ (the animal's oxygen supply capacity, *Seibel et al., 2021*). The gray bands represent the 95% confidence interval.

When both packages and their methods were compared (Table 3), only the $\alpha$-Pcrit method from the respirometry package produce significantly lower values (Nested Anova, Supplementary material 5). Hence, a DO of 3.7 ± 1.0 (average from all methods except

$\alpha$-Pcrit) could represent the lower bound $PO_2$ over which *P. capitata* corals can regulate their $MO_2$, and ambient DO values below this value could expose this coral to hypoxia. However, the limit to maintain an aerobic metabolism (independent of whether the animal is resting or at maximum activity) is 1.6 mg $O_2$ $L^{-1}$, the $\alpha$-Pcrit.

## DISCUSSION

### Oxygen conditions in reefs of Gorgona Island

Low dissolved oxygen (DO) concentration in coral reefs of Gorgona Island occurred during the upwelling season, and even hypoxic values ($<3.7$ mg $L^{-1}$) were recorded at 10 m depth. Additionally, DO varied significantly over years. Low DO concentrations during 2013 and 2022 coincided with the onset and full development (respectively) of La Niña conditions (*NOAA, 2022*), when winds and upwelling intensify (*Fiedler & Lavín, 2017*), while high DO concentration in 2016 and 2019 (Fig. 3) coincided with El Niño events (*Canchala et al., 2020*; *NOAA, 2022*) when the winds and upwelling are relaxed (*Ledesma et al., 2022*). These findings coincide with what has been previously described for the Colombian Pacific basin during different phases of ENSO events, with high DO conditions during an El Niño year (1998, 6.8 mg $L^{-1}$) and low DO conditions during a La Niña year (1999, 4.5 mg $L^{-1}$; *Devis-Morales et al., 2002*).

As we observed at Gorgona Island, at Matapalo reef (Gulf of Papagayo, Costa Rica) low DO (4.5 mg $L^{-1}$) occurs during extreme upwelling events at 3 m depth, but the water in this area is on average well oxygenated ($7.2 \pm 0.01$ mg $L^{-1}$ mean $\pm$ SE, 100–120% saturation; *Rixen, Jiménez & Cortés, 2012*; *Stuhldreier et al., 2015*). Likewise, in reefs at San Agustin Bay (Gulf of Tehuantepec, Mexico) and Taboguilla Island (Gulf of Panama) DO was lower during the upwelling season than during the non-upwelling season (Gulf of Tehuantepec: 3.3 *vs* 5.2 mg $L^{-1}$, Gulf of Panama: 3.6 *vs* 6.3 mg $L^{-1}$, *Ramírez-Gutiérrez et al., 2007*; *Lucey, Haskett & Collin, 2021*). However, none of these studies state if oxygen conditions in the Gulfs of Tehuantepec and Panama were statistically lower during the upwelling than the non-upwelling seasons.

The average DO conditions in coral reefs of Gorgona Island during the non-upwelling season (4.9 mg $L^{-1}$) are among the lowest in the ETP region and resemble the sporadically low oxygen conditions in the Gulf of Papagayo during the upwelling season. The lowest mean DO values in coral reefs of the ETP occur in the Gulf of Tehuantepec (3.3 mg $L^{-1}$), where upwelling events last longer (*D'Croz & O'Dea, 2007*; *O'Dea et al., 2012*). DO on coral reefs typically ranges from 3.4 to 13.6 mg $L^{-1}$ at 27 °C or 50% up to 200% air saturation (*Nelson & Altieri, 2019*), and hypoxic environments at 2.8 mg $L^{-1}$ or $<41\%$ air saturation (*Vaquer-Sunyer & Duarte, 2008*), which is in line with the normoxic ($4.6 \pm 0.89$ mg $L^{-1}$) and the hypoxic values that we reported for Gorgona (3.0–3.7 mg $L^{-1}$, Table 4).

Low oxygen water layers (from anoxia to $<1$ mg $L^{-1}$), between 50 m and 500 m water depth, and known as oxygen minimum zones, occur south of the equator (below the Peru Current) and off the coast of southern Mexico (below the Eastern Pacific warm pool), and during upwelling events reach sub-surface (20-50 m water depth) layers (*Fiedler & Talley, 2006*). During the past 50 years, the oxygen minimum zones have expanded vertically,

**Table 4  Dissolved oxygen concentration in different coral reefs of the world.**

| Site | Dissolved oxygen (mg L$^{-1}$) | Reference |
| --- | --- | --- |
| Eastern Tropical Pacific | | |
| *Non-Upwelling sites* | | |
|     Gulf of Chiriqui, Panama | 6.5 | *Camilli (2007)* |
|     Coco Island, Costa Rica | 5.3 | *Esquivel-Garrote et al. (2020)* |
|     Bahia de Navidad, Mexico | 7.2 | *Godínez-Domínguez et al. (2000)* |
| *Upwelling sites* | | |
|     Gulf of Tehuantepec, Mexico | 3.5–5.2 | *Ramírez-Gutiérrez et al. (2007)* |
|     Gulf of Papagayo, Costa Rica | 7.1–7.7 | *Stuhldreier et al. (2015)* |
|     Gulf of Panama, Panama | 3.6–6.3 | *Lucey, Haskett & Collin (2021)* |
|     Gorgona Island, Colombia | 4.2–4.9 | This study |
| Caribbean | | |
| *Non-Upwelling sites* | | |
|     Rosario Island, Colombia | 6 | *Severiche et al. (2017)* |
|     Bermuda. UK | 7.5 | *Ziegler et al. (2021)* |
|     Florida keys, USA | 6.5 | *Ziegler et al. (2021)* |
|     Puerto Rico | 6.4 | *Ziegler et al. (2021)* |
|     Cayo coral, Panama | 5.4 | *Johnson et al. (2021)* |
| *Upwelling sites* | | |
|     Tayrona, Colombia | 6.3 | *Bayraktarov, Pizarro & Wild (2014)* |
| Indopacific | | |
| *Non-Upwelling sites* | | |
| Cocos Islands, Australia | 4.5 | *Hobbs & McDonald (2010)* |
| Bouraké, New Caledonia | 5.5 | *Maggioni et al. (2021)* |
| Hawaii, USA | 6.8 | *Ziegler et al. (2021)* |
| *Upwelling sites* | | |
|     Nanwan Bay, Taiwan | 3.5–7.4 | *Meng et al. (2008)* |
| Red Sea | | |
| Aqaba, Jordan | 5.1–10.3 | *Wild et al. (2010)* |
| Gulf of Oman | 8.3 | *Bauman et al. (2010)* |
| **Mean DO** | 4.5–6.6 mg L$^{-1}$ | |

**Notes.**
Mean dissolved oxygen concentration (DO, mg L$^{-1}$) in different coral reef sites around the world. When more than one DO value is presented, the first corresponds to the upwelling season and the second to the non-upwelling season.

from a thickness of 370 m in 1960 to 690 m in 2006, and in the ETP there is a trend of oxygen loss of 49 mmol m$^2$ year$^{-1}$ (*Stramma et al., 2008*). It is likely that this tendency will continue, adding another potential threat to coral reef communities of the ETP (*Fiedler & Lavín, 2017*). Although the arrival of oxygen-poor subsurface water occurred only during the upwelling season at Gorgona Island, upwelling of hypoxic waters might become more common in the future due to the expansion of the oxygen minimum zone and a strengthening of the trade winds system (and its associated upwelling) due to climate change (*Rixen, Jiménez & Cortés, 2012*; *Bakun et al., 2015*; *Sydeman et al., 2014*; *Xiu et al., 2018*).

Fluctuations in DO are related to variation in the amount of organic matter, and the activity of aerobic organisms that decompose it (*Testa et al., 2014*). Oxygen minimum zones are the result of high phytoplanktonic production at the surface, a sharp permanent pycnocline that prevents local ventilation of subsurface waters, and a slow deep-water circulation (*Gooday et al., 2010*). We propose that the low DO concentration at Gorgona during the non-upwelling seasons, is due to high productivity that could increase the biological oxygen demand.

In the ETP high concentration of chlorophyll-a (Cl-a) occurs during the upwelling season. In the Gulf of Tehuantepec Cl-a concentration was 2.8 mg m$^{-3}$ during the upwelling season and 0.1 mg m$^{-3}$ during the non-upwelling season (*Coria-Monter et al., 2019*). In the Gulf of Papagayo, Cl-a concentration was 1.2 mg m$^{-3}$ during the upwelling season, and 0.59 mg m$^{-3}$ during the non-upwelling season (*Stuhldreier et al., 2015*). In the Gulf of Panama Cl-a was 1.4 and 0.2 mg m$^{-3}$ during the upwelling and non-upwelling seasons, respectively (*D'Croz & O'Dea, 2007*). At Gorgona Island Cl-a mean value on coral reefs was 5.0 mg m$^{-3}$ during the upwelling season, and 3.5 mg m$^{-3}$ during the non-upwelling season (A Giraldo, unpublished data, 2010, 2013, and 2018). Values of Cl-a concentrations at Gorgona Island are like those in Peru (5–2 mg m$^{-3}$, *Echevin et al., 2008*). Two factors lead to a high concentration of Cl-a throughout the year along the Colombian Pacific coast: first, upwelling throughout most of Panama Bight at the beginning of the year, and second, high pluviosity and numerous rivers on the region that supply large amounts of nutrients through runoff (*Corredor-Acosta et al., 2020*). Of local importance, Gorgona Island is 29 km offshore of the Sanquianga National Natural Park (NNP), where high sediment discharge from the Patia River occurs.

The Patia river (Fig. 1) has the largest delta on the western coast of South America (23,700 km$^2$), and its plume reaches Gorgona Island during La Niña events (*Restrepo & Kettner, 2012*). In 1972 a 3 km-long channel was constructed to connect the Patia and Sanquianga rivers. After that, more than 90% of the Patia River discharge started to flow northward through the Sanquianga River, leading to the degradation of the mangrove ecosystem in NNP Sanquianga (*Restrepo, 2012*; *Restrepo & Cantera, 2013*).

The actual sediment yield into the Pacific Ocean through the Sanquianga river is 1500 t km$^2$ yr$^{-1}$ (*Restrepo & Kettner, 2012*), and there is concern about the effects of sediment runoff on coral reefs of Gorgona because deforestation on the Colombian Pacific coast is increasing dramatically (*Belokurov et al., 2016*). Despite the importance of the Colombian Pacific forests in terms of carbon storage and as a biodiversity hot spot (*Canchala et al., 2020*), forest loss has been exacerbated in the region by illicit crops and alluvial mining to finance Colombia's armed conflict. Between 2001 and 2018 2324 km$^2$ of the Colombian Pacific Forest were deforested (*Anaya et al., 2020*), causing a 25.7% increase in deforestation of the mangrove at PNN Sanquianga (*Clerici et al., 2020*).

Eutrophication is considered a factor that induces hypoxia (*Diaz & Breitburg, 2009*). Nutrient excess from runoff increases the dominance of algae, which deplete oxygen during the night, smother coral by overgrowth, and in the longer term reduce coral cover and reef calcification (*Fabricius, 2005*). For example, runoff during floods results in massive mortality of coral reef organisms, after phytoplankton blooms deplete oxygen, together

with extremely low salinity and high nutrient concentration (*Jokiel et al., 1993*; *Lapointe & Matzie, 1996*; *Le Hénaff et al., 2019*). Examining the interactive effects of low salinity, high sediments, and low oxygen levels on scleractinian corals and establishing their tolerance is another important step to understand coral reef resilience at Gorgona Island.

## Hypoxic threshold of *Pocillopora capitata*

At 28 °C, the minimum level of $O_2$ required to sustain a constant $MO_2$ (0.26 ± 0.02 mg $O_2$ h$^{-1}$) of *Pocillopora capitata* was 3.7 mg L$^{-1}$ (56% air saturation). It is likely that below this DO, the aerobic metabolism of corals decrease, and could be exposed to hypoxia and undergo metabolic depression (*Reemeyer & Rees, 2019*). Protruding polyps, observed in *P. capitata* at the end of the experiment, is a behavior that might facilitate oxygen uptake (by increasing the surface in contact with the environment) during hypoxia (*Dodds et al., 2007*; *Henry & Torres, 2013*). Mucus production during the incubation could fuel microbial respiration, which also depletes $O_2$ (*Wild et al., 2010*). At Gorgona Island oxygen depletion due to mucus production could occur on the reef flat during spring low tides (*Castrillón-Cifuentes, Lozano-Cortés & Zapata, 2017*). Hence shallow corals could exhibit similar tolerance to hypoxic conditions to the ones studied here.

The hypoxic threshold of *P. capitata* ($P_{crit}$ 3.7 mg L$^{-1}$) and the low DO concentration recorded at La Azufrada reef (3.1 mg L$^{-1}$) and other coral reef sites of Gorgona Island (3.0–3.7 mg L$^{-1}$) evidenced that although these corals inhabit well-oxygenated waters (mean DO 4.6 ± 0.89 mg L$^{-1}$), they are close to the limit of hypoxic conditions, hence, when DO decreases from 3.7 mg L$^{-1}$ corals could face metabolic constraints depending on the duration of hypoxic conditions. Assessing the molecular responses to diel and seasonal changes in DO will help to understand the resilience of *Pocillopora* corals to deoxygenation (as proposed by *Murphy & Richmond, 2016*; *Zoccola et al., 2017*; *Alderdice et al., 2020*; *Alderdice et al., 2021*; *Deleja et al., 2022*).

A similar value between the $P_{crit}$ and the lower DO conditions on reefs were also found for 14 tropical scleractinian corals and the cold-water coral *Lophelia pertusa*. The $P_{crit}$ (at 26 °C, and 35 PSU) for the tropical corals range from 2 to 4 mg $O_2$ L$^{-1}$, and the lower ambient DO was 2 mg $O_2$ L$^{-1}$ (*Hughes et al., 2022*). The $P_{crit}$ (at 9 °C, and 35 PSU) for the cold-water coral was 3.9–4.3 mg $O_2$ L$^{-1}$, and the lower ambient DO 3.2 mL L$^{-1}$ (*Dodds et al., 2007*). Although there is a discussion around the use and way to calculate this metric (*Wood, 2018*; *Regan et al., 2019*), the similarity between $P_{crit}$ and the lowest DO ambient conditions highlight its ecological relevance, and in our case similar values of $P_{crit}$ were obtained from different methods.

Two Pocilloporids were included on *Hughes et al. (2022)* research, *Pocillopora acuta* ($P_{crit}$ *1.3* mg $O_2$ L$^{-1}$) and *Pocillopora damicornis* ($P_{crit}$ *1.2* mg $O_2$ L$^{-1}$). These congeners had $P_{crit}$ lower than the $P_{crit}$ of *P. capitata*, and it could be due to taxa, or a response to local changes in temperature and $O_2$ conditions (*Vaquer-Sunyer & Duarte, 2008*). However, $P_{crit}$ in *P. acuta* and *P. damicornis* were like the $\alpha$-$P_{crit}$ of *P. capitata* (1.6 mg $O_2$ L$^{-1}$), which is the oxygen limit to maintain the aerobic metabolism (*Seibel et al., 2021*).

We suspect that different Pocilloporids of the ETP could have different tolerance to hypoxic conditions, and due to thermal dependence of $P_{crit}$ (*Seibel et al., 2021*), even a

seasonal response could occur in *Pocillopora*, especially for corals that inhabit sites were upwelling develops (Table 4). Hence, we encourage researchers to assess the hypoxic tolerance of corals from the ETP, due to predictions of deoxygenation that the region will face (*Fiedler & Lavín, 2017*). However, trying to identify a species-specific response could be hampered due to taxonomic gaps for *Pocillopora* in the region.

At Gorgona Island DO values lower than 2.1 mg $L^{-1}$ occur at 30 m depth during the upwelling season (*Giraldo et al., 2014*). DO $<2$ mg $L^{-1}$ is considered a hypoxic condition because it is lethal for many aquatic organisms (*Levin et al., 2009*), but the lethal oxygen thresholds for aquatic animals are species-specific because they can range from 0.28 to 4 mg $L^{-1}$ (*Vaquer-Sunyer & Duarte, 2008*; *Welker et al., 2013*). The low DO values recorded in this study (3.0 mg $L^{-1}$) and by *Giraldo et al. (2014)* open the possibility that some benthic organisms (that occur deeper than 10 m) may be occasionally exposed to hypoxic conditions.

Because water samples to describe DO conditions were collected during daylight hours and were not regularly monitored, it is possible that even lower DO concentrations may occur at reefs of Gorgona Island. A study in the Red Sea found strong diurnal variation in $O_2$ concentrations (5.1 to 9.3 mg $L^{-1}$) in algae-dominated reefs compared to coral-dominated ones (6.6 to 8.5 mg $L^{-1}$), with the lowest DO values during dusk and before dawn, and highest values around midday (*Wild et al., 2010*). During the day, photosynthesis by zooxanthellae and benthic algae produce oxygen that exceeds the respiratory demands of reef aerobic organisms, but during the night photosynthetic processes stop and respiration of organisms depletes oxygen to levels at which severe hypoxic conditions (down to 0.7 mg $L^{-1}$) can occur (*Nelson & Altieri, 2019*).

We propose that the occurrence of hypoxic waters during the upwelling season at 20-30 m depth and even at 10 m depth during intense upwelling events, impose a bathymetric restriction for *Pocillopora* corals. In fact, the deepest distribution of scattered *Pocillopora* colonies at Gorgona Island is 20 m (*Muñoz, Jaramillo-González & Zapata, 2018*) and for coral reefs is ∼8 m (*Vargas-Ángel, 2003*). Untangling which abiotic factor(s) that reach extreme levels during upwelling events (*e.g.*, temperature, salinity, nutrients, pH or oxygen) limits the depth distribution of corals and coral reefs at Gorgona Island could be a potentially important line of research at this marine protected area. Most of the literature attributes the restriction of ETP coral reefs to shallow areas to low temperatures and low light conditions (*Glynn, Manzello & Enochs, 2017*). However, oxygen conditions play a major role in the success of benthic organisms, including calcification processes, a physiological function with key ecological consequences for reef accretion (*Wijgerde et al., 2012*; *Wijgerde et al., 2014*; *Nelson & Altieri, 2019*; *Hughes et al., 2020*; *Sampaio et al., 2021*).

When hypoxia is not lethal, it reduces growth, reproduction, photosynthesis, or immune responses due to low energy production in organisms, while the remaining energy is invested in survival (*Diaz, Rabalais & Breitburg, 2012*; *Dupont-Prinet et al., 2013*; *Wooldridge, 2014*; *Nelson & Altieri, 2019*; *Hughes et al., 2020*). Testing whether hypoxia induces trade-offs in *Pocillopora* corals (reducing vital functions like gamete production) to balance aerobic metabolism deprivation during upwelling events, could help to understand why *Pocillopora* in the ETP have low sexual reproduction (*Glynn et al., 1991*). A trade-off between growth

and sexual reproduction has already been observed in *P. damicornis* at Gorgona Island after stress due to aerial exposure during extreme low tides (*Castrillón-Cifuentes, Lozano-Cortés & Zapata, 2017*). Other trade-offs can also occur. For instance, the sea pen *Veretillum cynomorium* experiences oxygen deprivation in internal tissues during air exposure, but they display a costly enzymatic protective response to prevent tissue oxidation (*Teixeira et al., 2013*).

Other factors (independent of upwelling events) that potentially can induce hypoxia at coral reefs of the ETP are algal overgrowth, eutrophication, sedimentation (*Fabricius, 2005*; *Diaz & Breitburg, 2009*; *Murphy & Richmond, 2016*; *Nelson & Altieri, 2019*). For the ETP there are only two reports of *Pocillopora* mortality (at Caño Island, Costa Rica, and Utria, Colombia) apparently due to oxygen deprivation after a dinoflagellate bloom of *Chlodinium catenatum* and *Gonyaulax monilata* (*Guzman et al., 1990*; *Vargas-Ángel, 1996*).

## CONCLUSIONS

In conclusion, we found well-oxygenated water conditions in coral reefs of Gorgona Island, but noticed the occurrence of hypoxic waters reaching the deepest parts of reefs ($<3.7$ mg $O_2$ $L^{-1}$ at 10 m depth) during upwelling events. We found that the hypoxic threshold ($P_{crit}$) of *Pocillopora capitata* (3.7 mg $O_2$ $L^{-1}$) was similar to the low DO concentration recorded on coral reefs of Gorgona Island during the upwelling season at 10 m depth. Hence, if hypoxic events intensify because of climate change, or if local hypoxic inducing circumstances increase (*e.g.*, eutrophication from Patía River), this could represent a real threat to *Pocillopora* corals, the main reef-builder scleractinian coral at Gorgona Island. Preventing nutrient enrichment (that decreases water quality) and sediment input from the Patía River is an important task for the reef management at Gorgona Island. Further questions to solve are how long do hypoxic conditions persist at reefs of Gorgona, what is the frequency of occurrence of hypoxic events, how long can corals tolerate hypoxic conditions, the role of oxygen conditions on the bathymetric distribution of corals (especially in the calcification process), and what are the long-term effects of hypoxia on the physiology (the molecular response) of corals, including other branching and massive coral species different from *P. capitata* that inhabit Gorgona Island.

## ACKNOWLEDGEMENTS

We are grateful to Danna Velasco, Juan Fernando Rivera, Maria Paula Millan, and Ana Maria Millan members of the Coral Reef Ecology Research Group from Universidad del Valle, and to Bruce Hoyos, for their help during the fieldwork, and the staff of the National Natural Park of Gorgona for their support in coordinating all activities at the island. We thank the constructive comments from two anonymous reviewers.

### Funding

This work was funded by The Intergovernmental Panel on Climate Change and The Cuomo Foundation (through the IPCC Scholarship Program, 6th round), The Rufford Foundation small grant (31547-1), The PADI foundation (68128), and Colfuturo and Minciencias (202007248), who provided the funds for the doctoral research project of Ana Lucia Castrillón-Cifuentes. The University of Bremen provided the funds for the article publication. The contents are solely the liability of Ana Lucia Castrillón Cifuentes and under no circumstances may be considered as a reflection of the position of the funders. The funders had no role in study design, data collection and analysis, decision to publish, or preparation of the manuscript.

### Grant Disclosures

The following grant information was disclosed by the authors:
The Intergovernmental Panel on Climate Change and The Cuomo Foundation (through the IPCC Scholarship Program, 6th round).
The Rufford Foundation small grant: 31547-1.
The PADI foundation: 68128.
Colfuturo and Minciencias: 202007248.
The University of Bremen.

### Competing Interests

The authors declare there are no competing interests.

### Author Contributions

- Ana Lucia Castrillón-Cifuentes conceived and designed the experiments, performed the experiments, analyzed the data, prepared figures and/or tables, authored or reviewed drafts of the article, and approved the final draft.
- Fernando A. Zapata conceived and designed the experiments, analyzed the data, authored or reviewed drafts of the article, and approved the final draft.
- Alan Giraldo conceived and designed the experiments, performed the experiments, authored or reviewed drafts of the article, and approved the final draft.
- Christian Wild conceived and designed the experiments, authored or reviewed drafts of the article, and approved the final draft.

### Field Study Permissions

The following information was supplied relating to field study approvals (*i.e.*, approving body and any reference numbers):

The environmental authority of Parques Nacionales Naturales de Colombia endorsed all aims and methods of this research (PIR NO. 014.19, Agreement 239; December 20, 2019).

### Data Availability

The raw measurements are available in the Supplementary File.

## Supplemental Information

Supplemental information for this article can be found online at http://dx.doi.org/10.7717/peerj.14586#supplemental-information.

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
