# Peer review of "Spatiotemporal variability of oxygen concentration in coral reefs of Gorgona Island (Eastern Tropical Pacific) and its effect on the coral Pocillopora capitata"

_PeerJ, doi:10.7717/peerj.14586_

## Round 0.1 · original submission · Major Revisions

Both reviewers appreciate the work you've put into this manuscript, and have included a number of points to improve the manuscript. Please pay attention to the comments of Reviewer 1 in particular -- things that need to be improved are (1) background literature, (2) background and context around Pcrit, (3) add additional citations and quality assessment for the data sources used, (4) re-think the modeling approach used.

Reviewer 1 ·

Basic reporting

The manuscript is well-written overall with professional English used throughout. However, I believe the ms is lacking sufficient background/context and does not adequately cover the relevant references in the literature. The impact of hypoxia on corals is a relatively understudied area to date, and only a relatively small number of papers exist on this topic. So, I am surprised that most of these are overlooked - In fact, I would go as far as to say, the following refs are critical reading/reference material for a study concerned with coral hypoxia thresholds:

Alderdice et al. 2021 (Global Change Biology) – demonstrated a mechanistic link between hypoxia and coral bleaching response, and showed differential hypoxia tolerance among closely-related Acropora species.

Alderdice et al. 2022 – (Molecular Ecology) – shows susceptibility to hypoxia for corals across multiple life stages (i.e. larval and adult)

Haas et al. 2014 (PeerJ) – critical paper providing first data on coral hypoxia thresholds (and importantly, the first reported lethal threshold for an Acropora species)

Hughes et al. 2022 (Marine Pollution Bulletin) – to my knowledge the only work assessing oxyregulatory function of tropical scleractinians, which seems critical for work intending to use Pcrit (or cP) as an indicator of hypoxia tolerance/sensitivity

Johnson et al. 2021 (Scientific Reports) – critical paper providing data on coral hypoxia thresholds across multiple coral taxa

Deleja et al. 2022 (Biology) – highlights mechanisms of sub-lethal hypoxia on corals

This issue is further compounded by use of odd or inappropriate references to support statements/concepts within the text (e.g. Gravinense ref [line 70], which has nothing to do with land run-off – see also my specific comments below for further examples). Both the introduction and discussion would be greatly improved by incorporation of relevant references. N.B. There has also been a couple of papers from John Burt’s lab group (NYU Abu Dhabi) concerning expanding hypoxia in the Persian/Arabian Gulf region – again, I think incorporating such references into this paper would be beneficial/relevant.

Perhaps the most obvious area where the text could be improved by providing context is surrounding the concept of Pcrit (cP). This concept is not introduced (or indeed defined) in the manuscript – leaving the reader with no context as to what this value means, its interpretation, and any caveats surrounding its use to describe hypoxia tolerance (especially within organisms that unlikely function as “perfect” oxyregulators). There is heated debate playing out in the literature at present (see Wood 2020 vs. XX et al. 2021), and as such I believe the authors would significantly improve their manuscript by providing a clear definition of Cp (and some physiological context), together with a justification for their use of this descriptor to hypoxia tolerance. I’m not saying the authors shouldn’t use Cp, but as there are several approaches to analysing oxygen consumption patterns, it is important to provide the reader with a rationale for selection/use of this metric.

Experimental design

The authors address relevant and meaningful questions - in a nutshell: how variable is DO in and around their study area, and how do these vales relate to physiological hypoxia thresholds of a main reef-forming coral found there.

A major issue however is that most DO data were extracted from technical reports (i.e. not performed by the authors themselves) – but no references are provided for these data sources (is it a database? or multiple written technical reports?). This makes it impossible to assess the rigour/quality of the measurements performed (e.g., how often was the DO probe calibrated?, how was the calibration on the DO probe verified after calibrating in water-saturated air?, how many replicate measurements were taken for each sample? How were samples handled between Niskin deployment and eventual measurement with the DO probe?, what units were DO originally measured in, - % air saturation or mg/L?). The latter question is a concern because the units of DO partial pressure (% air saturation) and DO concentration (mg/L) cannot be reconciled later in the manuscript. For example, on line 197, the authors state that in their incubation chamber (with seawater of 30 PSU and 28C) a DO concentration of 6.7 mg L-1 corresponds to a partial pressure of 84% air saturation. This is incorrect, as an O2-solubility table will confirm that 6.7 mg L-1 is actually >100 % air saturation. So one (or both) units are incorrect here, which leads me to wonder whether similar issues affect the data extracted from reports? (i.e., have the authors inter-converted between DO units?).

A standard check of calibration accuracy when using hand-held DO probes (e.g. the YSI), or the portable Pyroscience kit (used in the lab component) is to verify the calibration against an O2-solubility table above, with reference seawater of a known salinity and temperature that has been fully-aerated. It is not clear that this was performed in either the field or lab – and if it was performed in the lab, how was the discrepancy between partial pressure and concentration not captured? (Perhaps the authors can go back and verify the unit conversion/update the text).

With regards to the assessment of hypoxia thresholds, the methodology to collect O2-consumption patterns is routine (aside from the DO unit issue) - whereby the authors use respirometry to measure DO drawdown over time with sufficient acclimation time at the start of the incubation. The fact that only a single coral species was investigated, and that incubations were terminated after a pre-determined time (regardless of DO concentration) are potential problems however, particularly with the way the analyses have been done (see next review section).

Validity of the findings

Unfortunately there are some shortcomings in the analysis of the hypoxia threshold(s). Firstly, is Pcrit an appropriate statistic to describe hypoxia tolerance in corals? - The relevance of Pcrit is being debated even for known strong oxyregulating taxa such as fish (see "feud" between Mark Wood and Seibel in the literature) - so how valid is it for an organism that is likely only moderately capable of regulation? (I think the ms would benefit greatly from justification/supporting text). Regardless, of whether Pcrit is appropriate or not, a major weakness here is that the authors plot all VO2 data points (i.e. for all 7 coral fragments) on a single graph, and then fitting a single model (a 2-segment linear fit), to yield only a single value for PCrit (i.e. effectively n=1) with no average or SE. Typically when analysing O2 consumption curves for organisms (whether fish or corals), the PO2 vs VO2 plot for each individual replicate is fitted with a model and Pcrit extracted from each one - so the authors can explore the variance in across their data. Eyeballing their data clearly shows the model is a very poor fit to the combined data, so I think the authors might want to rethink this approach. Some justification for the arbitrary location of the two linear segments is also required (I don't see believe the provided Seibel ref is adequate justification). A potential issue i see here however, is that the incubations have been terminated at the same time regardless of DO consumption - and therefore some replicates will lack data points at lower PO2 values for which to reliably fit with the 2-part model (and perhaps is why the data have been lumped together?). It would be far more robust to analyse each oxygen consumption curve separately and I would strongly encourage the authors to take that approach. Finally, my only other observation is that without knowing how Pcrit actually relates to hypoxia stress (whether sub-lethal or lethal) - what can be informed from knowing it for a single species in the study area? (some discussion is surely needed here).

Additional comments

Specific Comments (line number provided)

54 – Surely there are many more appropriate references for this statement (i.e. foundational data papers by e.g., Terry Hughes, Ove Hoegh-Guldberg)

70 – Not clear how the Gravinense et al. 2022 ref is appropriate here? (it has nothing to do with land-based run-off).

122 – so the O2 sensor was calibrated with a single point calibration in water-saturated air? – was this calibration verified afterwards, and if so, how?

181 – I don’t understand how the DO concentration is 9.7 mg L at 30 PSU and 28 C – at 100% air saturation, the DO concentration should be XX mg L – so why the starting levels so high?

194 – Water can not be supersaturated with O2 from using an air pump (you can bubble air all day into seawater and you will achieve 100% air saturation). The only way you can achieve a DO > 100% air saturation is if you introduce additional O2 into the water (by either photosynthesis or though direct O2 injection).

197 – again, at 28C (and here I assume 30 PSU) – 100% air saturation would be ~6.5 mg/L – is this an issue with the calibration or unit conversion?

264 – the concept of oxyregulation has not been introduced at all up to this point – and yet this is ultimately the critical metric by which the hypoxia threshold of the coral is determined (i.e. where oxyregulation apparently stops, and oxyconformity ensues). There isn’t even a definition or reference relating to oxyregulation anywhere in the text.

282 – how are you defining hypoxia here?

309 – Altieri and Nelson do not state that hypoxia for corals appears at <41 % - they simply state that 41% is likely above the hypoxia threshold for many benthic species. It would be more appropriate here to compare the cP (Pcrit) values against actual measured hypoxia threshold values for corals (there are a few – e.g., , Haas et al. 2014; Johnson et al. 2022; Hughes et al. 2022).

380 – I am not sure that mucus production would facilitate O2 uptake – if anything wouldn’t it further restrict the diffusion of O2 between the coral and surrounding seawater? (also I don’t believe this is evidenced by either of the two refs provided here?).

Cp critical oxygen partial pressure point – is this the same as PCrit? This is not described at all in the manuscript – what is it? What does it mean from a physiological basis? How does it compare to values for other organisms? There isn’t even discussion of whether corals are oxyconformers or oxyregulators to even determine whether Pcrit even exists for a coral?

Fig 6 inflection point of the curve – is this curve showing data from all coral fragments on a single graph? Why have the data been collectively visualised like this?

Reviewer 2 ·

Basic reporting

1. BASIC REPORTING: The manuscript is written using clear English. However, some minor suggestions are added at the end of the review that might improve readability. The introduction offers enough background to understand the topic
Data: It is not clear if the raw data and data analyses are available and how. I think the authors have not made it available.
Figures: Fig 1. Please make the red and white points bigger. It is hard to find them on the maps. Fig 1C is not mentioned until the end of the manuscript. I am not sure if this figure is needed for the paper. If the authors want to keep it, I would suggest mentioning it in the text when the rest of figure 1 is described.
Fig 5. It is confusing that the variability units of the data in panels A and B are different from C. Please make them consistent or make figure 5C its own figure.
Fig 6. Please describe what the dots and the multiple lines are in the figure legend. Describe what the acclimation section means.
Figures 5 and 6 represent the hypothesis tested and I would suggest the authors emphasize these. I found figure 2 distracting and I did not understand its relevance to the paper.

Experimental design

2. EXPERIMENTAL DESIGN
The research problem is defined and the study fills a knowledge gap regarding the oxygen concentrations around Gorgona reefs and the physiological effects of oxygen concentration on a common coral species in the area. The methods are described in detail.

Validity of the findings

3. VALIDITY OF THE FINDINGS
Data reported in figure 4 is confusing when compared with the findings in figure 5. Significant differences were found when complete levels were sampled in the years 2013, 2017 and 2019 (figure 5), but in figure 4 the authors plot the mean values of unbalanced treatment levels. I am not sure this is the best way of finding the true mean for the reefs, depths, years… etc. because the means are weighed towards the conditions that were sampled more times.
My other major concern with the experimental design is that only 7 corals were used in the physiological experiment. The authors should justify how this is representative of the possible coral responses.
The authors should still make the data available.
It is not clear the specific factors' interactions and nesting design in the ANOVA. Please include a table with the complete ANOVA outputs (at least as a supplement).

Additional comments

Some general comments
Abstract: Line 22-24. These two sentences are not supported by the literature in the introduction. The authors should either support them or present them as hypotheses instead of affirmations
Lines 24-27: This objective is confusing. It is not clear how field data helps to understand the effects of oxygen variability on the corals. Perhaps the experimental phase could be used to understand the effects on corals because it actually measures a physiological response? Or the objective should be to understand the effects of oxygen variability on the Gorgona Reefs? The authors better phrased the goals in lines 94-97.
Materials & Methods: Lines 137-143: I am not sure about the value of this analysis or figure 5. Given that the data is imbalanced, pooling sites, depths, and seasons to calculate mean values for a specific target might be very biased for the type of data more available for that respective group. I would consider at least making panels A and B one panel where you can have the mean by year for non-upwelling and upwelling, and panels C and D in one panel showing the means but site at 1 and 10m.
Lines 154-160: It would be worth mentioning why La Azufrada reef was sampled only during the upwelling season and not during the non-upwelling. The physiological data from La Azufrada was collected during the non-upwelling. Could this lead to differences in how they respond to hypoxia? Is it possible that the corals experience some acclimation to seasonal conditions?
Line 164: Would you expect these corals to be acclimated to lower oxygen than shallower corals?
Line 174: Figure 2 seems unnecessary for the methods and does not describe the results discussed. Maybe it could be added as a supplement? Seven colonies is quite a low N for an experiment.
Line 208-210: What are the circles in figure 3? Not sure what pattern could be seen. If this paragraph refers to figure 6, what are the 2 lines describing?
Results: I am not sure the results from Lines 227-241 and figure 4 are relevant. The mean values (I am happy to be corrected here) are imbalanced. There are no statistical differences among reefs, depths, or seasons, but this is different from years 2013, 2017 and 2019 where seasons and years are significant factors.
265-267: I do not see this acclimation trend in figure 6.

Minor suggestions for the authors to consider:

Introduction: Line 29: The mean concentration in GORGONA IS. coral reefs? (add Gorgona Is.)
Line 43: Oxygen CONCENTRATION? Is one of the main …
Line 58: RCPs are not used anymore. They are noW SSPs.
Line 60: Remove “,” after et al.?
Line 83: Please add the variability metric that is presented with this mean value. Is it SD? SE?
Line 92: There is no definition of micro-environment (the spatial scale referred to here) and it is not clear why this is important for the study.
Line 97: one of the main reef-building species IN THE ETP REGION?

Materials & Methods

Line 109: which corals cope at (add? ETP) reefs
Line 109: Since 2005 the program HAS ASSESSED
Line 115-117: The sentence reads a bit weird with the placement of the commas. Please consider rearranging.
Lines 114-115: It seems unnecessary to describe the distribution of the 24 monitoring stations since only 5 are used. It would be simpler to describe the 5 stations used in the manuscript (Lines 127 and after).
Lines 129: Moving parenthesis in 129 after “Gorgona Island” (line 130) could improve readability.
Line 295: What are the variability units? SD? SE?
Multiple lines. Be consistent with Gorgona island abbreviation. Sometimes it is used and sometimes it is fully spelled.

334: What are the Ch-a values? What’s the reference for this sentence?

---

## Round 0.2 · accepted · Accept

Thanks for your work addressing the previous reviewers' comments, all are happy with the revision. Congratulations!

Reviewer 1 ·

Basic reporting

Is written reasonably well, however there are a number of typos throughout and I suggest the authors do a thorough proof-reading of the text before publication. e.g. (here is what I came across in the intro inside 2 mins of reading:

61 - remove gap in P O2
66 – organisms
78 – change leaded to led
79 – Haas not Hass

Beyond this, I did not read critically for typos/errors - so I would strongly suggest that the authors go through this text carefully to eliminate any remaining.

Experimental design

no comment - authors have addressed the concerns listed in my previous review

Validity of the findings

no comment (as per my previous review)

Additional comments

The authors have made a good effort to address issues raised in my previous review and I believe the ms has been improved as a result.

A very minor thing: I would remove the abbreviation of Island (Is.) - it’s not easy on the eye, and it's not clear why the abbreviation is suddenly in the methods, when the full word has been used throughout the introduction. I suggest just using the full word throughout